# MITIGATING PARAMETER INTERFERENCE IN MODEL MERGING VIA SHARPNESS-AWARE FINE-TUNING

**Yeoreum Lee**[1]**, Jinwook Jung**[1] **& Sungyong Baik**[1,2†]
[1] Dept. of Artificial Intelligence, [2] Dept. of Data Science
Hanyang University
{leeyeoreum01,jjw970517,dsybaik}@hanyang.ac.kr

## ABSTRACT

Large-scale deep learning models with a pretraining-finetuning paradigm have led to a surge of numerous task-specific models fine-tuned from a common pre-trained model. Recently, several research efforts have been made on merging these large models into a single multi-task model, particularly with simple arithmetic on parameters. Such merging methodology faces a central challenge: interference between model parameters fine-tuned on different tasks. Few recent works have focused on designing a new fine-tuning scheme that can lead to small parameter interference, however at the cost of the performance of each task-specific fine-tuned model and thereby limiting that of a merged model. To improve the performance of a merged model, we note that a fine-tuning scheme should aim for (1) smaller parameter interference and (2) better performance of each fine-tuned model on the corresponding task. In this work, we aim to design a new fine-tuning objective function to work towards these two goals. In the course of this process, we find such objective function to be strikingly similar to sharpness-aware minimization (SAM) objective function, which aims to achieve generalization by finding flat minima. Drawing upon our observation, we propose to fine-tune pre-trained models via sharpness-aware minimization. The experimental and theoretical results showcase the effectiveness and orthogonality of our proposed approach, improving performance upon various merging and fine-tuning methods. Our code is available at https://github.com/baiklab/SAFT-Merge.

## 1 INTRODUCTION

Foundation model, a large deep learning model pre-trained on large-scale datasets, has shown great advancement across a wide range of downstream tasks, after fine-tuning on each task (Achiam et al., 2023; Saab et al., 2024; Ding et al., 2023). Recent successes of the pretraining-finetuning paradigm have given rise to a burst of task-specific open-source models in communities, such as Hugging Face. Diversity yet ready availability of large task-specific models have naturally elicited a question from researchers: Can we combine these large models into one, while retaining the performance on each task?

Traditionally, a single multi-task model is obtained by jointly training on data across all tasks (Caruana, 1997; Crawshaw, 2020; Vandenhende et al., 2022). However, given the size of foundation models and the number of tasks, joint training on all tasks incurs significant computational costs. Motivated by the accessibility, variety, abundance, and common origin of task-specific models, several research efforts have focused on merging multiple fine-tuned models into a single model via simple arithmetic on parameters of these models, thereby removing the need for joint training (Ilharco et al., 2023; Yadav et al., 2023; Yang et al., 2024b; Matena & Raffel, 2022; Jin et al., 2023; Daheim et al., 2024; Li et al., 2023; Yang et al., 2024a). However, a central challenge remains: parameters of different task-specific models interfere or conflict with each other, leading to the performance degradation of a merged multi-task model on each task.

To bridge such performance gap, several works have tried to reduce the parameter interference during the process of merging (Yadav et al., 2023; Jin et al., 2023; Yang et al., 2024b; Wang et al.,

---

[†]Corresponding author.

2024; Yu et al., 2024). Another line of works focuses on finding a new fine-tuning scheme that results in task-specific models whose parameters have lower parameter interference (also often referred to as better weight disentanglement with respect to model outputs) (Ortiz-Jimenez et al., 2023; Tang et al., 2024; Jin et al., 2025) and thus less performance degradation after merging. Few studies (Wortsman et al., 2022a; Ilharco et al., 2023; Wortsman et al., 2022b) suggest that the effectiveness of linear arithmetic on parameters in the process of merging may be owed to the linearity of fine-tuning process. Conversely, Ortiz-Jimenez et al. (2023) have refuted such hypothesis by showing that there is a huge performance drop from the approximation of fine-tuned models with a linearized pre-trained model. Another observation they make is that such post-hoc linearized models led to less parameter interference. Based on this observation, few recent works Ortiz-Jimenez et al. (2023); Tang et al. (2024); Jin et al. (2025) have tried to explicitly linearize fine-tuning processes in order to induce weight disentanglement.

In this work, we note that we need to simultaneously work towards two goals for effective model merging: (1) *reducing parameter interference between fine-tuned models* while (2) *maintaining the performance of task-specific fine-tuned models on respective datasets*. Therefore, during fine-tuning process, we aim to directly optimize for both performance on each task and weight disentanglement with respect to performance. In the course of designing a fine-tuning objective function that aligns with our goals, we find striking resemblances between our goals and sharpness-aware minimization (SAM) (Foret et al., 2021), which aims for better generalization by finding flat minima via minimization of both loss values and loss sharpness. In particular, we find the similarities between the minimization of both loss values and loss sharpness in SAM and joint optimization for performance and weight disentanglement of fine-tuned models in our goal.

Drawing upon our observations, we propose to obtain task-specific models from pre-trained models via sharpness-aware fine-tuning (SAFT), in order to achieve better performance on each task, lower parameter interference, and thus better overall performance of a merged multi-task model. Our extensive experimental results demonstrate that our proposal greatly improves the overall performance of a merged model. The effectiveness of our proposed method is owed to achieving better performance of each task-specific model and less performance gap between task-specific models and a merged model. We further highlight the generalizability and orthogonality of our approach by demonstrating performance improvements when applied together with various merging methods and fine-tuning methods for model merging.

## 2 RELATED WORKS

**Model merging.** The recent emergence of large foundation models and pretraining-finetuning paradigm has motivated researchers to explore ways of merging multiple task-specific models into a single multi-task model without re-training. Model merging, the merging of models with simple arithmetic on parameters, has garnered a significant amount of attention for its flexibility and simplicity. However, parameters of different task-specific models may interfere with each other during merging process, resulting in performance degradation on each task, compared to task-specific models. To address the parameter interference issue, researchers focus on either designing a merging process (Utans, 1996; Ilharco et al., 2023; Yadav et al., 2023) or designing a fine-tuning process to mitigate the parameter interference. Initiated with simple averaging (Utans, 1996; Shoemake, 1985), research works on merging process focus on representing task-specific models as task vectors for easier manipulation of knowledge (Ilharco et al., 2023), or weighting parameters (Matena & Raffel, 2022; Jin et al., 2023; Yang et al., 2024b) or selecting parameters (Yadav et al., 2023; Wang et al., 2024; Yu et al., 2024) according to the estimated importance of each parameter with respect to given tasks. In parallel, if task-specific model parameters have less interference with each other to begin with, the effectiveness of model merging can be amplified. As such, few recent works have focused on designing a fine-tuning process such that resulting fine-tuned model parameters will have less interference and result in less performance gap between a merged model and task-specific models. Ortiz-Jimenez et al. (2023) show that linearized fine-tuning (fine-tuning in the space tangent to pre-trained initialization) leads to less interference (specifically better weight disentanglement with respect to model outputs), aspring other linearized fine-tuning methods (Jin et al., 2025; Tang et al., 2024).

**Sharpness-aware minimization (SAM).** Foret et al. (2021) introduce a new optimization objective function that minimizes both loss and loss sharpness to seek flat loss minima that may lead to better generalization performance. SAM defines loss sharpness as the maximum loss difference measured at current parameters and nearby parameters (obtained by perturbing current parameters). Several follow-up works have strived to improve SAM via improving perturbation methods (Mi et al., 2022; Kwon et al., 2021), improving gradient (Wang et al., 2023; Zhao et al., 2022), or combining with other flatness-aware optimizers (Cha et al., 2021; Kaddour et al., 2022) for better generalization. While previous studies have primarily focused on single-task learning, Phan et al. (2022) incorporate SAM into joint multi-task training as a regularization technique for multi-task learning. We note that our method and Phan et al. (2022) target two different scenarios. Phan et al. (2022) target a traditional multi-task learning scenario, where the training is performed on all tasks jointly. By contrast, our work tackles multi-task model merging, where the goal is to merge different task-specific models, each of which is independently fine-tuned from a common pre-trained model without the knowledge of other tasks. This approach eliminates the need for joint training on all tasks at the same time and avoids retraining from scratch when new tasks are introduced. In multi-task model merging, the lack of knowledge of other tasks also presents several challenges, such as parameter interference between different task-specific models that cause degradation of single-task performance after merging.

In this work, we introduce a new objective function for single-task fine-tuning aimed for model merging, from which we present a new insight that draws connections between the objective of multi-task model merging and that of sharpness-aware minimization (SAM). Furthermore, we theoretically (in Appendix D) and empirically show that, sharpness-aware fine-tuning can reduce parameter interference, even without the knowledge of other tasks during fine-tuning.

## 3 BACKGROUND

**Sharpness-aware minimization (SAM).** To achieve better generalization, SAM (Foret et al., 2021) seeks for wider minima by minimizing both loss value and loss sharpness during optimization, where the loss sharpness is formulated as a difference between a loss at the current parameters and the maximum loss value at nearby parameter values:

$$\min_{\boldsymbol{\theta}} \left[ \underbrace{\max_{\boldsymbol{\epsilon}:\|\boldsymbol{\epsilon}\|_2 \leq \rho} \mathcal{L}(\boldsymbol{\theta} + \boldsymbol{\epsilon}; \mathcal{D}) - \mathcal{L}(\boldsymbol{\theta}; \mathcal{D})}_{\text{loss sharpness}} \right] + \underbrace{\mathcal{L}(\boldsymbol{\theta}; \mathcal{D})}_{\text{loss}}, \tag{1}$$

where $\boldsymbol{\epsilon}$ is a perturbation vector which is bounded above by a predefined $\rho$ that controls the radius of the neighborhood; and $\boldsymbol{\theta}$ are network parameters to be optimized for a given loss function $\mathcal{L}$ over a dataset $\mathcal{D}$. For efficiency, Foret et al. (2021) approximate the inner maximization via Taylor approximation. Then, along with canceling identical terms $\mathcal{L}(\boldsymbol{\theta}; \mathcal{D})$ with opposite signs, the original optimization is reduced to

$$\min_{\boldsymbol{\theta}} \mathcal{L}(\boldsymbol{\theta} + \hat{\boldsymbol{\epsilon}}; \mathcal{D}) \quad \text{where} \quad \hat{\boldsymbol{\epsilon}} \triangleq \rho \frac{\nabla_{\boldsymbol{\theta}} \mathcal{L}(\boldsymbol{\theta}; \mathcal{D})}{\|\nabla_{\boldsymbol{\theta}} \mathcal{L}(\boldsymbol{\theta}; \mathcal{D})\|}. \tag{2}$$

The perturbations within the same neighborhood radius for all parameters may impact each parameter differently, especially if their scales differ by several factors. To take such varying scales of parameters into account, Adaptive SAM (ASAM) (Kwon et al., 2021) proposes to scale the perturbation vector $\boldsymbol{\epsilon}$ according to the scale of each parameter as $\hat{\boldsymbol{\epsilon}}_{\text{ASAM}} \triangleq \rho \frac{\boldsymbol{\theta}^2 \nabla_{\boldsymbol{\theta}} \mathcal{L}(\boldsymbol{\theta}; \mathcal{D})}{\|\nabla_{\boldsymbol{\theta}} \mathcal{L}(\boldsymbol{\theta}; \mathcal{D})\|}$. Adjusting the scale of perturbations according that of parameters can be even more effective in large foundation models with the pretraining-finetuning paradigm, since large pre-trained models likely have a large number of parameters of different scales after training on large-scale datasets.

**Problem setting.** In the pretraining-finetuning paradigm, there exists a large pre-trained model $f : \mathcal{X} \times \Theta \rightarrow \mathcal{Y}$, parameterized by trained parameters $\boldsymbol{\theta}_0 \in \Theta$, that is in turn fine-tuned to $T$ downstream tasks. Each downstream task, indexed by $t$, is accompanied with a dataset $\mathcal{D}^{(t)} = \{(\boldsymbol{x}_i^{(t)}, y_i^{(t)})\}_{i=1}^{N_t}$, where $\boldsymbol{x}_i^{(t)} \in X^{(t)} \subseteq \mathcal{X}$ is an input with a corresponding label $y_i^{(t)} \in Y^{(t)} \subseteq \mathcal{Y}$. Employing a standard loss function (e.g., cross-entropy loss for classification) and an optimizer (e.g., SGD), fine-tuning a pre-trained model $f_{\boldsymbol{\theta}_0}$ to each downstream task $t$ will lead to a task-specific model $f_{\boldsymbol{\theta}_t}$ with

its parameters $\boldsymbol{\theta}_t$:

$$\boldsymbol{\theta}_t = \arg\min_{\boldsymbol{\theta}} \mathcal{L}(\boldsymbol{\theta}; \mathcal{D}^{(t)}). \tag{3}$$

**Task arithmetic.** To merge models into a single model by performing an arithmetic on model parameters, Ilharco et al. (2023) have introduced the concept of task vector, which is essentially a vector pointing to task-specific parameters $\boldsymbol{\theta}_t$ from pre-trained model parameters $\boldsymbol{\theta}_0$, obtained by taking a difference between them: $\boldsymbol{\tau}_t = \boldsymbol{\theta}_t - \boldsymbol{\theta}_0$. Ilharco et al. (2023) note that a task vector $\boldsymbol{\tau}_t$ can be considered as the representation of the knowledge learned for a task $t$. As such, they claim that the knowledge of each task can be manipulated by a simple arithmetic on pre-trained model parameters: $\boldsymbol{\theta}_0 + \alpha_t \boldsymbol{\theta}_t$, where $\alpha_t > 0$ will add the knowledge of task $t$ while $\alpha_t < 0$ will result in forgetting the knowledge of task $t$, while $|\alpha_t|$ controls the extent of learning/forgetting. Using these task vectors $\boldsymbol{\tau}_t$ with corresponding task coefficients $\alpha_t$, task-specific models can be merged into a merged multi-task model, parameterized by $\boldsymbol{\theta}_{\mathrm{merge}}$ as follows:

$$\boldsymbol{\theta}_{\mathrm{merge}} = \boldsymbol{\theta}_0 + \sum_{t=1}^{T} \alpha_t \boldsymbol{\tau}_t. \tag{4}$$

## 4 MITIGATING PARAMETER INTERFERENCE VIA SHARPNESS-AWARE FINE-TUNING

Since a merged model is formed by simply performing linear arithmetic on task vectors, there is a high chance for interference among tasks (Ilharco et al., 2023). Such interference leads to the performance degradation on downstream tasks after merging. Some works focus on reducing interference during merging process, which is a challenging task as fine-tuned model parameters are fixed. On the other hand, few recent works propose to modify a fine-tuning process that results in task-specific models whose parameters have less interference with each other. In particular, they show that fine-tuning a (partially) linearized model or only its linear layers results in less interference. However, such linearization of fine-tuning process results in the performance degradation of each task-specific model, limiting the overall performance of a merged model.

In this work, we claim that we need to achieve both (1) *less performance gap between a merged model and each fine-tuned model (i.e., less parameter interference)* and (2) *generalization performance of each fine-tuned model* on each respective dataset. As such, we aim to design a new objective function for fine-tuning to achieve these two objectives:

$$\boldsymbol{\theta}_t = \arg\min_{\boldsymbol{\theta}} \underbrace{\mathcal{L}(\boldsymbol{\theta}_{\mathrm{merge}}(\boldsymbol{\theta}); \mathcal{D}^{(t)}) - \mathcal{L}(\boldsymbol{\theta}; \mathcal{D}^{(t)})}_{\text{Objective (1)}} + \underbrace{\mathcal{L}(\boldsymbol{\theta}; \mathcal{D}^{(t)})}_{\text{Objective (2)}}, \tag{5}$$

where $\boldsymbol{\theta}_{\mathrm{merge}}(\boldsymbol{\theta})$ is to demonstrate that $\boldsymbol{\theta}_{\mathrm{merge}}$ changes as $\boldsymbol{\theta}$ is optimized, while considering parameters for other tasks to be fixed. While this objective function already looks similar to the SAM objective function in Equation 1, after some simplifications (deriviations are delineated in Appendix B), we get the final objective function as follows:

$$\boldsymbol{\theta}_t = \arg\min_{\boldsymbol{\theta}} \mathcal{L}(\boldsymbol{\theta} + \sum_{s \neq t} \alpha_s \boldsymbol{\tau}_s + (\alpha_t - 1)\boldsymbol{\tau}; \mathcal{D}^{(t)}), \tag{6}$$

where $\sum_{s \neq t} \alpha_s \boldsymbol{\tau}_s + (\alpha_t - 1)\boldsymbol{\tau}$ represents the parameter offsets a model merging process would introduce to the parameters of a task-specific model for a task $t$. Hence, $\sum_{s \neq t} \alpha_s \boldsymbol{\tau}_s + (\alpha_t - 1)\boldsymbol{\tau}$ can be considered as perturbations that would cause parameter interference during model merging, from the perspective of each task-specific model. However, we do not assume access to other tasks while fine-tuning on each task, as each task-specific model is independently trained. Since other tasks are unknown (and thus $\sum_{s \neq t} \alpha_s \boldsymbol{\tau}_s$ is unknown), we consider $\sum_{s \neq t} \alpha_s \boldsymbol{\tau}_s + (\alpha_t - 1)\boldsymbol{\tau}$ to be random perturbations. Furthermore, because the perturbation $\sum_{s \neq t} \alpha_s \boldsymbol{\tau}_s + (\alpha_t - 1)\boldsymbol{\tau}$ depends on $\boldsymbol{\tau} = \boldsymbol{\theta} - \boldsymbol{\theta}_0$ and is thus varying during training, we use ASAM that models $\hat{\epsilon}$ as parameter-dependent perturbation. In other words, we use $\hat{\epsilon}_{\mathrm{ASAM}}$ as a surrogate of $\sum_{s \neq t} \alpha_s \boldsymbol{\tau}_s + (\alpha_t - 1)\boldsymbol{\tau}$, approximating our objective function for fine-tuning aimed for model merging (Equation 5) as

$$\min_{\boldsymbol{\theta}} \mathcal{L}(\boldsymbol{\theta} + \hat{\epsilon}; \mathcal{D}) \quad \text{where} \quad \hat{\epsilon} = \rho \frac{\boldsymbol{\theta}^2 \nabla_{\boldsymbol{\theta}} \mathcal{L}(\boldsymbol{\theta}; \mathcal{D})}{\|\nabla_{\boldsymbol{\theta}} \mathcal{L}(\boldsymbol{\theta}; \mathcal{D})\|}. \tag{7}$$

From our perspective described above, we can consider parameter interference to be caused by parameter perturbations $\sum_{s \neq t} \alpha_s \boldsymbol{\tau}_s + (\alpha_t - 1)\boldsymbol{\tau}$ that would be introduced during model merging, the information of which is however not available during fine-tuning for each task. The perturbations will bring a model to a new location in the loss landscape, away from the found local minimum. If the region around the local minimum is not flat enough, the new location (i.e., merged model parameters) brought by perturbations will most likely have a higher loss, resulting in a large performance gap between a merged model and a task-specific model. In other words, to minimize the interference caused by parameter perturbations, it is essential to identify flat minima. Flat minima can effectively prevent the loss from increasing after parameter perturbations (e.g., model merging). Thus, we argue that finding flat minima (or equivalently, minimizing sharpness) via sharpness-aware fine-tuning (SAFT) can greatly reduce parameter interference.

## 5 EMPIRICAL AND THEORETICAL ANALYSIS

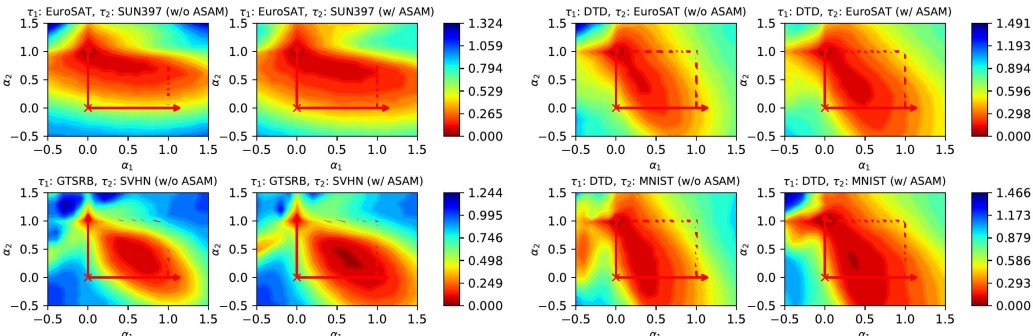

Figure 1: **Weight disentanglement visualization of two task-specific models across two tasks.** Each pixel in the heatmap corresponds to the weight disentanglement error $\xi(\alpha_1, \alpha_2)$ between a two-task-merged model, parameterized by $\boldsymbol{\theta}_{\text{merge}} = \boldsymbol{\theta}_0 + \alpha_1 \boldsymbol{\tau}_1 + \alpha_2 \boldsymbol{\tau}_2$, and two task-specific models, parameterized by $\boldsymbol{\theta}_0 + \alpha_1 \boldsymbol{\tau}_1$ and $\boldsymbol{\theta}_0 + \alpha_2 \boldsymbol{\tau}_2$, evaluated on task 1 and task 2. We use CLIP ViT-B/32 on EuroSAT-SUN397, DTD-EuroSAT, GTSRB-SVHN, and DTD-MNIST task pairs to plot these visualizations. The red box highlights the search space used to find the optimal task coefficients $\{\alpha_1, \alpha_2\}$ for task arithmetic.

In this section, we experimentally validate our argument by showing that our proposal, SAFT, leads to better weight disentanglement (Figure 1 and Figure 2), better cross-task linearity (Figure 3), and better joint-task loss linearity (Figure 4 and Figure 5), which are the signs of less parameter interference. Better performance by our proposed method, compared to standard SGD and other fine-tuning schemes specifically designed for model merging, further underlines the effectiveness of SAFT in reducing parameter interference. Then, we also theoretically show that the capability of SAFT to reduce the dominant Hessian eigenvalues induces joint-task loss linearity (the linearity of loss on all joint tasks).

**Weight disentanglement.** Ortiz-Jimenez et al. (2023) argue that for model merging via task arithmetic to be effective, weight disentanglement (a task vector for one task not affecting the outputs of task-specific model on other tasks) is a necessary condition. In this work, we show that SAFT indeed achieves better weight disentanglement, in comparison to a standard objective function. The weight disentanglement is achieved when the parameter updates with the task vector for the task $t$ (i.e., $\boldsymbol{\tau}_t$) impart a localized influence on the output of a model only when processing an input from the task $t$, without impacting the output of the model when inputs from other tasks are processed. Ortiz-Jimenez et al. (2023) formally express the localized influence of task vectors on the input space as

$$f(\boldsymbol{x}; \boldsymbol{\theta}_{\text{merge}}) = f\left(\boldsymbol{x}; \boldsymbol{\theta}_0 + \sum_{s=1}^{T} \alpha_s \boldsymbol{\tau}_s\right)$$

$$= f(\boldsymbol{x}; \boldsymbol{\theta}_0 + \alpha_t \boldsymbol{\tau}_t) \quad \text{when} \quad \boldsymbol{x} \in X^{(t)}. \tag{8}$$

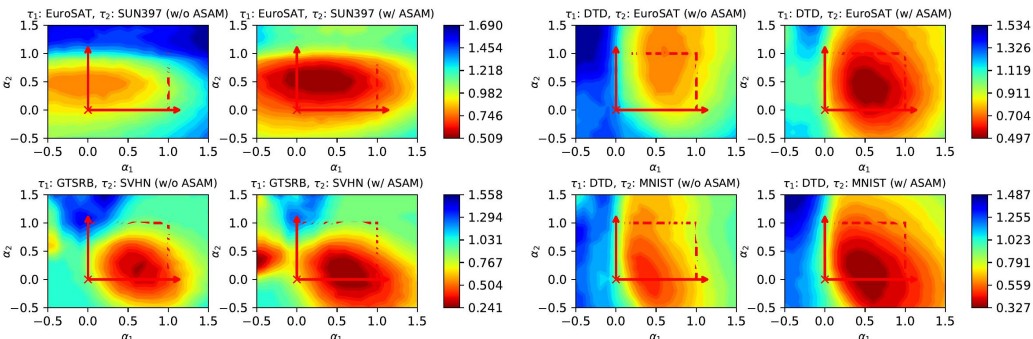

Figure 2: **Weight disentanglement visualization of eight-task-merged models across two tasks**. Each pixel in the heatmap corresponds to the disentanglement error $\xi(\alpha_1, \alpha_2)$ between an eight-task-merged model, parameterized by $\boldsymbol{\theta}_{\text{merge}} = \boldsymbol{\theta}_0 + \alpha_1 \boldsymbol{\tau}_1 + \alpha_2 \boldsymbol{\tau}_2 + \sum_{s \notin \{1,2\}} \alpha_s \boldsymbol{\tau}_s$, and each task-specific model, evaluated on task 1 and 2. To visualize the landscape of the merged multi-task model on a 2D heatmap, we adjust only two task coefficients corresponding to the evaluation tasks. The models and evaluation task pairs used for the visualization are the same as those used in Figure 1. The meaning of the red box is the same as in Figure 1.

To evaluate how well weight disentanglement is satisfied, Ortiz-Jimenez et al. (2023) quantify disentanglement error as the discrepancy between the output of a merged model and $t$-th task-specific model on input data of $t$-th task. Lower disentanglement errors imply that each task contributes appropriately without adversely affecting others. Following the experimental settings by Ortiz-Jimenez et al. (2023), we first evaluate weight disentanglement during the merging of two tasks:

$$\xi(\alpha_1, \alpha_2) = \sum_{t=1}^{2} \mathbb{E}_{\boldsymbol{x} \in X^{(t)}} \left[ \text{dist} \left( f(\boldsymbol{x}; \boldsymbol{\theta}_0 + \alpha_t \boldsymbol{\tau}_t), f(\boldsymbol{x}; \boldsymbol{\theta}_0 + \alpha_1 \boldsymbol{\tau}_1 + \alpha_2 \boldsymbol{\tau}_2) \right) \right], \tag{9}$$

where $\xi(\alpha_1, \alpha_2)$ is the disentanglement error with respect to two given tasks and visualized in Figure 1. We further stress-test and evaluate the disentanglement error while considering merging of all task-specific models ($T = 8$ in this work), thereby evaluating how well an actual merged model achieves weight disentanglement. However, it is difficult to visualize if all $T$ task coefficients are adjusted. In this work, for ease of visualization, we adjust task-coefficients of two tasks while fixing other task coefficients, while still considering a merged model with all task vectors:

$$\xi_{\text{all}}(\alpha_1, \alpha_2) =$$
$$\sum_{t=1}^{2} \mathbb{E}_{\boldsymbol{x} \in X^{(t)}} \left[ \text{dist} \left( f(\boldsymbol{x}; \boldsymbol{\theta}_0 + \alpha_t \boldsymbol{\tau}_t), f(\boldsymbol{x}; \boldsymbol{\theta}_0 + \alpha_1 \boldsymbol{\tau}_1 + \alpha_2 \boldsymbol{\tau}_2 + \sum_{s \notin \{1,2\}} \alpha_s \boldsymbol{\tau}_s) \right) \right], \tag{10}$$

where $\xi_{\text{all}}(\alpha_1, \alpha_2)$ represents the total disentanglement error across all tasks (visualized in Figure 2, and $\text{dist}(\cdot, \cdot)$ is a distance metric measuring the divergence between the outputs of the task-specific model and the merged model. Small $\xi(\alpha_1, \alpha_2)$ or $\xi_{\text{all}}(\alpha_1, \alpha_2)$ implies that the merged model parameter $\boldsymbol{\theta}_{\text{merge}}$ better reflects the individual contribution of each task, signifying reduced parameter interference. Indeed, the visualizations of weight disentanglement when considering two tasks in Figure 1 and all tasks ($T = 8$) in Figure 2 demonstrate the effectiveness of our method in achieving better weight disentanglement. In particular, we note that the weight disentanglement error of the model merging with standard fine-tuning optimization increases significantly when considering all tasks in model merging, compared to considering two tasks. On the other hand, our proposal, SAFT, reduces the weight disentanglement even when considering all tasks, further highlighting the effectiveness of our method in model merging.

**Cross-task linearity.** Cross-Task Linearity (CTL) (Zhou et al., 2024) is a property that ensures the linear separability of task influences on the layer outputs across all layers of the network. To satisfy CTL, for every layer $\ell$, the layer output of a merged model should be approximately equal to the combination of the layer outputs of individual task-specific models scaled by their respective coefficients. Formally, Zhou et al. (2024) define CTL condition as:

$$f^{(\ell)}(\boldsymbol{x}; \lambda \boldsymbol{\theta}_s + (1 - \lambda) \boldsymbol{\theta}_t) \approx \lambda f^{(\ell)}(\boldsymbol{x}; \boldsymbol{\theta}_s) + (1 - \lambda) f^{(\ell)}(\boldsymbol{x}; \boldsymbol{\theta}_t), \tag{11}$$

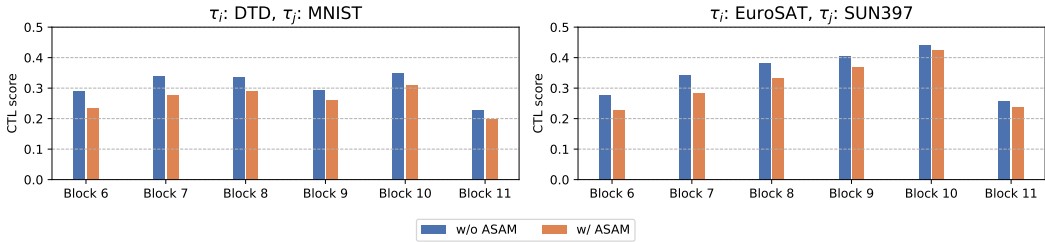

Figure 3: **Verification of CTL between the merged model and task-specific models**. We compare $\mathbb{E}_{\mathcal{D}^{(s)} \cup \mathcal{D}^{(t)}}[1 - \cos^{(\ell)}(\boldsymbol{x}; 2\lambda\boldsymbol{\tau}_s, 2\lambda\boldsymbol{\tau}_t)]$ between sharpness-aware fine-tuning and SGD. The values for the last six blocks are evaluated on the two task pairs DTD-MNIST and EuroSAT-SUN397. We set the scaling factor $\lambda$ to $0.3$.

where $\lambda \in \mathbb{R}$ is a scaling factor; $\boldsymbol{x} \in X^{(s)} \cup X^{(t)}$ is an input from either task $s$ or $t$; $\boldsymbol{\theta}_s, \boldsymbol{\theta}_t$ are parameters of task-specific models fine-tuned on task $s$ and $t$ respectively; and $f^{(\ell)}(\boldsymbol{x}; \boldsymbol{\theta})$ represents the response (or a feature) of $\ell$-th layer of a network $f$ for the given input $\boldsymbol{x}$. The cross-task linearity at each layer implies that the influence of one task on another is minimal, thereby facilitating effective weight disentanglement. Zhou et al. (2024) demonstrate that satisfying CTL condition leads to reducing the disentanglement error $\xi(\alpha)$. To evaluate whether CTL is satisfied, the following cosine similarity metric is used:

$$
\begin{aligned}
&\cos^{(\ell)}(\boldsymbol{x}; 2\lambda\boldsymbol{\tau}_s, 2\lambda\boldsymbol{\tau}_t) \\
&= \cos\left[ f^{(\ell)}(\boldsymbol{x}; \boldsymbol{\theta}_0 + \lambda(\boldsymbol{\tau}_s + \boldsymbol{\tau}_t)), \frac{1}{2}f^{(\ell)}(\boldsymbol{x}; \boldsymbol{\theta}_0 + 2\lambda\boldsymbol{\tau}_s) + \frac{1}{2}f^{(\ell)}(\boldsymbol{x}; \boldsymbol{\theta}_0 + 2\lambda\boldsymbol{\tau}_t) \right].
\end{aligned} \tag{12}
$$

The metric measures the cosine similarity between the layer output of a merged model and the averaged layer outputs of the task-specific models. Following the settings in (Zhou et al., 2024), we use the metric $\mathbb{E}_{\mathcal{D}}[1 - \cos^{(\ell)}(\boldsymbol{x}; 2\lambda\boldsymbol{\tau}_s, 2\lambda\boldsymbol{\tau}_t)]$ to evaluate how well CTL is satisfied, where smaller values of $\mathbb{E}_{\mathcal{D}}[1 - \cos^{(\ell)}(\boldsymbol{x}; 2\lambda\boldsymbol{\tau}_s, 2\lambda\boldsymbol{\tau}_t)]$ indicate stronger CTL. Since satisfying CTL leads to better weight disentanglement, smaller values of $\mathbb{E}_{\mathcal{D}}[1 - \cos^{(\ell)}(\boldsymbol{x}; 2\lambda\boldsymbol{\tau}_s, 2\lambda\boldsymbol{\tau}_t)]$ should result in lower disentanglement error $\xi(\alpha_1, \alpha_2)$, as noted by Zhou et al. (2024). Figure 3 shows that our method has lower $\mathbb{E}_{\mathcal{D}}[1 - \cos^{(\ell)}(\boldsymbol{x}; 2\lambda\boldsymbol{\tau}_s, 2\lambda\boldsymbol{\tau}_t)]$ in comparison to SGD, demonstrating that our method results in not just better weight disentanglement, but also better cross-task linearity.

**Joint-task loss landscape.** We empirically demonstrate that our method reduces parameter interference by finding flatter minima across the joint tasks. Figure 4 shows the joint-task loss landscape visualizations for two task-specific models trained on corresponding two tasks. We observe that our method allows models to reach flatter minima across the joint tasks compared to SGD, particularly around the boundaries of the task coefficients search space in task arithmetic. Our method increases the likelihood of finding a merged model connected to each task-specific model along a low-loss path, which indicates a smaller performance gap between the merged model and the task-specific models. Consequently, our proposal makes it easier to find a merged model with reduced parameter interference, compared to SGD.

Yet, the tendencies observed with two tasks may not generalize in the case of more tasks, as the chance and extent of interference increases with the number of tasks (Yadav et al., 2023). Therefore, we also visualize the loss landscape of the multi-task model built by merging all eight tasks, parameterized by $\boldsymbol{\theta}_{\mathrm{merge}} = \boldsymbol{\theta}_0 + \sum_{t=1}^{8} \alpha_t \boldsymbol{\tau}_t$. To visualize the loss landscape of an eight-task-merged model on a 2D heatmap, we vary only the coefficients of two randomly chosen tasks.

Figure 5 shows the loss landscape of a multi-task model obtained by merging the parameters of eight task-specific models. Compared to Figure 4, the minima in the landscape shrink in every case as the number of task-specific models to be merged increases. However, while the minima found by SGD shrink significantly, the minima found by our method remain to be flat and wide in all cases. This suggests that our method maintains the ability to reduce parameter interference and preserve the performance of the merged model, even when more task-specific models are merged.

**Theoretical results.** Here, we further validate our proposal by theoretically demonstrating that SAFT leads to the loss of a merged model and task-specific models being connected along a linear

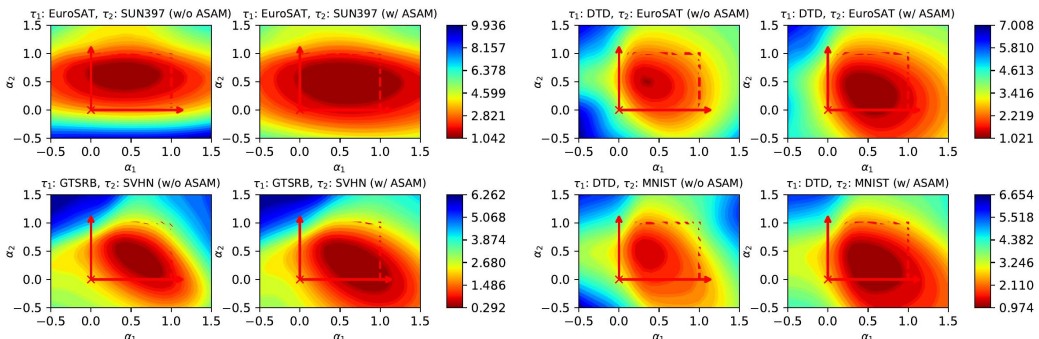

Figure 4: **Joint-task loss landscape visualization of two task-specific models across two tasks**. Each pixel in the heatmap corresponds to the loss values $\mathcal{L}(\boldsymbol{\theta}_{\mathrm{merge}}; \mathcal{D}^{(1)}) + \mathcal{L}(\boldsymbol{\theta}_{\mathrm{merge}}; \mathcal{D}^{(2)})$ of the two-task-merged model, parameterized by $\boldsymbol{\theta}_{\mathrm{merge}} = \boldsymbol{\theta}_0 + \alpha_1\boldsymbol{\tau}_1 + \alpha_2\boldsymbol{\tau}_2$, evaluated on task 1 and task 2. The setting of the model, task pairs, and red box is the same as in Figure 1. We use CLIP ViT-B/32 on the EuroSAT-SUN397 and DTD-MNIST task pairs to plot these visualizations. The red box highlights the search space used to find the optimal task coefficients $\{\alpha_1, \alpha_2\}$ of task arithmetic.

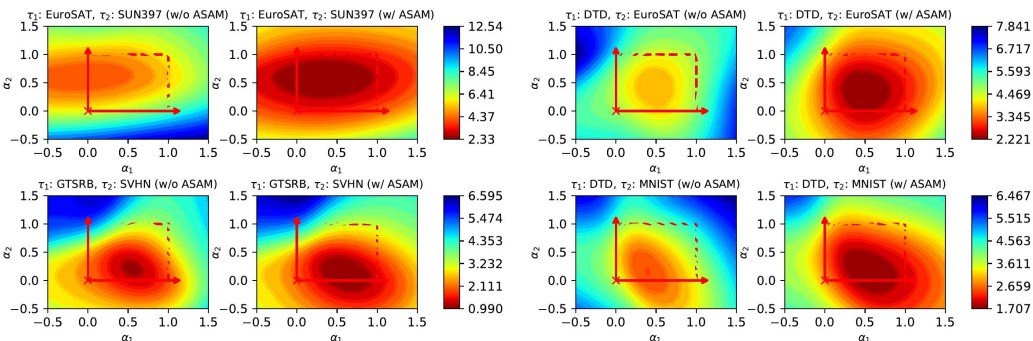

Figure 5: **Joint-task loss landscape visualization of eight-task-merged models across two tasks**. Each pixel in the heatmap corresponds to the loss values $\mathcal{L}(\boldsymbol{\theta}_{\mathrm{merge}}; \mathcal{D}^{(1)}) + \mathcal{L}(\boldsymbol{\theta}_{\mathrm{merge}}; \mathcal{D}^{(2)})$ of the eight-task-merged model, parameterized by $\boldsymbol{\theta}_{\mathrm{merge}} = \boldsymbol{\theta}_0 + \sum_{t=1}^{8} \alpha_t\boldsymbol{\tau}_t$, evaluated on tasks 1 and 2. We adjust only the two task coefficients corresponding to the evaluation tasks to visualize the weight disentanglement on a 2D map, as in Figure 2. The setting of the model, task pairs, and red box are the same as in Figure 4. We use the same models and task pairs as illustrated in Figure 4.

path on the loss landscape over all tasks. Formally, we define the loss over all tasks as joint-task loss and models being connected along a linear path on the joint-task loss landscape as joint-task loss linearity as follows:

**Definition 1** (Joint-task loss). *Given a joint-task dataset $\mathcal{D} = \mathcal{D}_s \cup \mathcal{D}_t$ and a model with parameters $\boldsymbol{\theta}$, we define Joint-Task Loss, denoted as $\mathcal{L}_{JTL}(\boldsymbol{\theta}; \mathcal{D})$, as:*

$$\mathcal{L}_{JTL}(\boldsymbol{\theta}; \mathcal{D}) = \mathcal{L}(\boldsymbol{\theta}; \mathcal{D}_s) + \mathcal{L}(\boldsymbol{\theta}; \mathcal{D}_t), \tag{13}$$

*where $\mathcal{L}(\boldsymbol{\theta}; \mathcal{D}_s)$ and $\mathcal{L}(\boldsymbol{\theta}; \mathcal{D}_t)$ denote the losses incurred by a model, parameterized by $\boldsymbol{\theta}$ on datasets $\mathcal{D}_s$ and $\mathcal{D}_t$, respectively.*

**Definition 2** (Joint-task loss linearity). ***Joint-task loss linearity (JTL linearity)*** *describes the linear relationship between the joint-task loss of an interpolated model and the weighted sum of the individual joint-task losses of task-specific models. Specifically, given models, parameterized by $\boldsymbol{\theta}_s$ and $\boldsymbol{\theta}_t$ fine-tuned on respective tasks with datasets $\mathcal{D}_s$ and $\mathcal{D}_t$, we say JTL linearity holds over a joint-task dataset $\mathcal{D} = \mathcal{D}_s \cup \mathcal{D}_t$ if*

$$\mathcal{L}_{JTL}(\alpha\boldsymbol{\theta}_s + (1-\alpha)\boldsymbol{\theta}_t; \mathcal{D}) \approx \alpha\mathcal{L}_{JTL}(\boldsymbol{\theta}_s; \mathcal{D}) + (1-\alpha)\mathcal{L}_{JTL}(\boldsymbol{\theta}_t; \mathcal{D}), \tag{14}$$

*where $\alpha \in [0, 1]$ is a scalar.*

**Theorem 1** (SAFT induces joint-task loss linearity). *If models, parameterized by $\boldsymbol{\theta}_s$ and $\boldsymbol{\theta}_t$, are obtained by fine-tuning from a common pre-trained model via SAFT on their respective datasets, the models better satisfy the joint-task loss linearity. A proof is relegated to Appendix D due to space constraints.*

Joint-task loss linearity induced by SAFT implies that SAFT leads to a merged model with less performance degradation in comparison to task-specific models on the joint task, thus implying that models fine-tuned by our method experience less parameter interference during the process of model merging.

## 6 EXPERIMENTS

In this section, following the settings from Ortiz-Jimenez et al. (2023), we conduct experiments on diverse tasks to demonstrate the effectiveness of our proposal, sharpness-aware fine-tuning (SAFT), in improving the overall performance of a merged model. We compare against three fine-tuning approaches: SGD, linearized fine-tuning in the tangent space (FTTS) (Ortiz-Jimenez et al., 2023), and fine-tuning linear layers only (FTLO) (Jin et al., 2025). We also validate the effectiveness, applicability, and generalizability of SAFT by assessing its performance in combination with three different model merging methods: weight averaging, task arithmetic (Ilharco et al., 2023), and TIES (Yadav et al., 2023).

### 6.1 TRAINING SETUP

Following the same training protocol outlined in Ilharco et al. (2022), we fine-tune two CLIP (Radford et al., 2021) models: (a) ViT-B/32 and (b) ViT-B/16. Our experiments are conducted across eight diverse datasets: (1) Cars (Krause et al., 2013), (2) DTD (Cimpoi et al., 2014), (3) EuroSAT (Helber et al., 2019), (4) GTSRB (Stallkamp et al., 2011), (5) MNIST (Deng, 2012), (6) RESISC45 (Cheng et al., 2017), (7) SUN397 (Xiao et al., 2016), (8) SVHN (Netzer et al., 2011). All fine-tuning processes begin from the same CLIP pre-trained checkpoint obtained from the `open_clip` (Radford et al., 2021) repository. We fine-tune each model for 8000 iterations with a batch size of 128 and a learning rate of $10^{-5}$ for all backbones and all fine-tuning methods. The learning rate schedule follows a cosine annealing approach with 500 warm-up steps, and optimization is performed using the AdamW (Loshchilov & Hutter, 2019). Consistent with Ilharco et al. (2022), we freeze the weights of the classification layer derived from encoding a standard set of zero-shot template prompts for each dataset. This strategy ensures that no additional learnable parameters are introduced during fine-tuning and does not compromise model accuracy. For more experimental details, please refer to Appendix A.

### 6.2 MAIN RESULTS

We evaluate the effectiveness of SAFT in closing the performance gap between a merged model and each task-specific models, in comparison to other three fine-tuning approaches. Table 1 shows that SAFT achieves the higher absolute and normalized accuracies in every case, compared to other fine-tuning methods. Normalized accuracy is defined as the absolute accuracy divided by the corresponding accuracy of the fine-tuned task-specific model, evaluating the performance gap between a merged model and task-specific models. These results suggest that SAFT not only improves performance in downstream tasks but also narrows the performance gap between the merged model and fine-tuned models, improving the overall performance. Moreover, SAFT can be used together with other fine-tuning methods (FTTS (Ortiz-Jimenez et al., 2023) and FTLO (Jin et al., 2025)), enhancing the performance in multi-task settings during model merging, demonstrating its generalizability and applicability. In particular, FTTS (Ortiz-Jimenez et al., 2023) and FTLO (Jin et al., 2025) demonstrate better multi-task performance compared to SGD, as these fine-tuning methods reduce interference between tasks by encouraging weight disentanglement (Malladi et al., 2023; Ortiz-Jimenez et al., 2023). Thus, the performance improvement brought by SAFT on top of these fine-tuning methods demonstrates the orthogonality of our proposal.

In Table 2, our method is shown to bring consistent performance improvement across diverse model merging methods and image encoder model backbones. Notably, our method brings performance improvement when used together with both weight averaging and task arithmetic. Task arithmetic

Table 1: **Multi-task performance across different fine-tuning methods**. We report the average absolute and normalized accuracies for three fine-tuning baselines: SGD, FTTS, and FTLO. Results are shown for three fine-tuning methods, grouped by whether SAFT-ASAM is applied. ViT-B/32 is used as the image encoder of CLIP, with task arithmetic as the model merging method in every case across eight tasks.

| Fine-tuning method ($\rightarrow$) | SGD | | FTTS | | FTLO | |
| --- | --- | --- | --- | --- | --- | --- |
| | Abs. | Norm. | Abs. | Norm. | Abs. | Norm. |
| w/o SAFT-ASAM | 68.23 | 75.47 | 78.35 | 86.83 | 75.93 | 85.74 |
| w/ SAFT-ASAM (Ours) | **69.45** | **76.32** | **79.38** | **87.72** | **77.49** | **88.77** |

Table 2: **Multi-task performance across different model merging methods and image encoder models**. We report the average absolute and normalized accuracies for three model merging methods: weight averaging, task arithmetic, and TIES merging. We also compare the performance of two different models used as the image encoder of CLIP: ViT-B/32 and ViT-B/16. All cases are fine-tuned using SGD and evaluated across eight tasks.

| Merging method ($\rightarrow$) | Weight averaging | | Task arithmetic | | TIES merging | |
| --- | --- | --- | --- | --- | --- | --- |
| | Abs. | Norm. | Abs. | Norm. | Abs. | Norm. |
| | ViT-B/32 | | | | | |
| SGD | 65.72 | 72.91 | 68.23 | 75.47 | 74.57 | 82.29 |
| SAFT-ASAM (Ours) | **66.76** | **73.62** | **69.45** | **76.32** | **75.45** | **82.86** |
| | ViT-B/16 | | | | | |
| SGD | 71.58 | 77.37 | 73.40 | 79.31 | 77.94 | 84.04 |
| SAFT-ASAM (Ours) | **71.84** | **77.53** | **76.77** | **82.50** | **80.14** | **86.23** |

searches for the optimal task coefficient within a given search space, while weight averaging is a specific case of task arithmetic, where $\alpha_t = \frac{1}{T}$. This suggests that our method finds flatter minima that covers the task coefficients search space in task arithmetic compared to SGD. Moreover, SAFT also performs better in the case of TIES merging, indicating that interference mitigation by our method complements the parameter interference mitigation achieved by TIES merging.

## 7 CONCLUSION

In this work, we draw connections between two research fields of machine learning: sharpness-aware minimization and multi-task model merging. Particularly, the connections are drawn from the formulation of two objectives of model merging: (1) reducing parameter interference between task-specific models and (2) achieving better generalization of each task-specific model. Building upon the objectives of model merging, we derive a new objective function for fine-tuning, from which we find similarities with sharpness-aware minimization. Upon observation, we propose to approximate our newly derived fine-tuning objective function with sharpness-aware minimization: sharpness-aware fine-tuning (SAFT). Experimental and theoretical results demonstrate that SAFT indeed results in less performance interference and better performance of a merged model, even when applied together with other merging and fine-tuning methods designed for model merging. Motivated by the effectiveness and applicability of our proposal, we hope that this work encourages further research on investigating the relationship between SAFT and model merging, opening a new research avenue.

ACKNOWLEDGMENTS

This research was supported by Basic Science Research Program through the National Research Foundation of Korea(NRF) funded by the Ministry of Education (No.RS-2024-00393305) and Institute of Information & communications Technology Planning & Evaluation (IITP) grant funded by the Korea government (MSIT) (No.RS-2020-II201373, Artificial Intelligence Graduate School Program (Hanyang University)).

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

## A    EXPERIMENTAL DETAILS

### A.1    FINE-TUNING BASELINES

We compare the following three fine-tuning baselines with and without SAFT:

(1) **SGD**: This refers to standard fine-tuning that uses only an optimizer such as AdamW (Loshchilov & Hutter, 2019).

(2) **F**ine-**T**uning in the **T**angent **S**pace (**FTTS**)  (Ortiz-Jimenez et al., 2023): This fine-tunes the model in the tangent space at its pre-trained initialization. It achieves this by linearizing the model using a first-order Taylor expansion $f_{\mathrm{lin}}(\boldsymbol{\theta}; \mathcal{D}) = f(\boldsymbol{\theta}_0; \mathcal{D}) + (\boldsymbol{\theta} - \boldsymbol{\theta}_0)^\top \nabla f(\boldsymbol{\theta}_0; \mathcal{D})$, where $\boldsymbol{\theta}_0$ represents the parameters of the pre-trained model and $\mathcal{D}$ is the training dataset. The method freezes $\boldsymbol{\theta}_0$ and updates only $\boldsymbol{\theta}$.

(3) **F**ine-**T**uning **L**inear Layers **O**nly (**FTLO**) (Jin et al., 2025): This exclusively fine-tunes the linear layers within the attention module.  Therefore, this method can only be applied to model architectures that include attention modules such as Transformer (Vaswani et al., 2017).

We utilize ASAM (Kwon et al., 2021) as a default SAFT method in every experiments, since it finds minima adaptively by considering correlation between generalization gap and sharpness. We set the $\rho$ value of ASAM to $0.5$, following the default setup outlined in ASAM, along with all other ASAM hyperparameters.

### A.2    MERGING METHODS

We merge the models that achieve the best performance for each corresponding task.  These best models are selected based on their performance on a validation set split, which is split from the training set at a $0.1$ ratio, as specified in Ilharco et al. (2023).

We use the following model merging methods as baselines:

(1) **Weight averaging**: This merges fine-tuned models by averaging their parameters element-wise, denoted as $\boldsymbol{\theta}_{\mathrm{merge}} = \frac{1}{T} \sum_{t=1}^{T} \boldsymbol{\theta}_t$, where $\boldsymbol{\theta}_t$ represents the fine-tuned parameters for each corresponding downstream task, $T$ is the number of downstream tasks being merged.

(2) **Task arithmetic** (Ilharco et al., 2023): This method calculates task vectors $\boldsymbol{\tau}_t = \boldsymbol{\theta}_t - \boldsymbol{\theta}_0$ for each downstream task $t$, where $\boldsymbol{\theta}_t$ represents the fine-tuned parameters for task $t$ and $\boldsymbol{\theta}_0$ represents the pre-trained parameters.  A linear combination of these task vectors is then added to the pre-trained parameters, denoted as $\boldsymbol{\theta}_{\mathrm{merge}} = \boldsymbol{\theta}_0 + \sum_{t=1}^{T} \alpha_t \boldsymbol{\tau}_t$, where $\alpha_t$ is a task coefficient that scales the corresponding task vector.  This method generalizes the weight averaging when $\alpha_t = \frac{1}{T}$ for $t = 1, 2, \ldots, T$.

Since the search space for $\alpha_t$ becomes too large as the number of tasks increases, we set the task coefficients to be the same for all tasks and search for the optimal coefficient within the range $[0.1, 0.3, 0.5, 0.7, 0.9, 1.0]$ using the validation set of each task.

(3) **TIES merging** (Yadav et al., 2023): This method mitigates parameter interference before merging models.  First, it trims parameters changed that change minimally during fine-tuning, as these small changes in each model can become more pronounced after element-wise parameter merging. Second, it resolves parameter interference due to sign conflicts by determining the sign of each parameter through a majority election before merging the models.

We apply TIES merging to task arithmetic. To find the optimal merged model, we search for the task coefficients in task arithmetic within the range $[0.1, 0.3, 0.5, 0.7, 0.9, 1.0]$ and the percentile of task vectors to be pruned to zero within $[0.7, 0.8, 0.9]$, using the validation set for each task.

### A.3    VISUALIZATION SETUP

We produce the joint-task loss landscape and disentanglement error under two distinct settings: (1) merging two task-specific models across two tasks, and (2) merging all eight task-specific models across two tasks.

1. Two task-specific models across two tasks: In the first setting, we parameterize the merged model as:
$$\boldsymbol{\theta}_{\text{merge}} = \boldsymbol{\theta}_0 + \alpha_1 \boldsymbol{\tau}_1 + \alpha_2 \boldsymbol{\tau}_2,$$
where $\boldsymbol{\tau}_1$ and $\boldsymbol{\tau}_2$ represent the parameters of models fine-tuned on tasks 1 and 2, respectively.

2. The eight-task-merged model across two Tasks: In the second setting, we parameterize the merged model as:
$$\boldsymbol{\theta}_{\text{merge}} = \boldsymbol{\theta}_0 + \alpha_1 \boldsymbol{\tau}_1 + \alpha_2 \boldsymbol{\tau}_2 + \sum_{s \notin \{1,2\}} \alpha_s \boldsymbol{\tau}_s,$$
with $\alpha = 0.3$, where $\boldsymbol{\tau}_s$ denotes the parameters of the additional six tasks.

For both settings, we use $(\alpha_1, \alpha_2)$ pairs spanning from $-0.5$ to $1.5$ with 21 evenly spaced points along each axis, resulting in a $21 \times 21$ grid. For each $(\alpha_1, \alpha_2)$ task coefficient pair on the grid, we compute the disentanglement error $\xi(\alpha_1, \alpha_2)$ and visualize the error values using contour plots to identify regions where the weight disentanglement is stronger. Since task coefficients are real numbers, we utilize contour plots to effectively visualize the variations in loss landscape and disentanglement error across the continuous $(\alpha_1, \alpha_2)$ parameter space.

## B    DERIVATION OF EQUATION 6

We start with simplifying Equation 5, which is the objective function that incorporates the goals of model merging:
$$\boldsymbol{\theta}_t = \arg \min_{\boldsymbol{\theta}} \mathcal{L}(\boldsymbol{\theta}_{\text{merge}}(\boldsymbol{\theta}); \mathcal{D}^{(t)}) - \mathcal{L}(\boldsymbol{\theta}; \mathcal{D}^{(t)}) + \mathcal{L}(\boldsymbol{\theta}; \mathcal{D}^{(t)})$$
$$= \arg \min_{\boldsymbol{\theta}} \mathcal{L}(\boldsymbol{\theta}_{\text{merge}}(\boldsymbol{\theta}); \mathcal{D}^{(t)}).$$

Here, we consider task coefficients $\{\alpha_s\}$ and other task vectors $\{\boldsymbol{\tau}_s\}_{s \neq t}$ to be fixed. Since then, instead of $\boldsymbol{\theta}_{\text{merge}} = \boldsymbol{\theta}_0 + \sum_{s=1}^{T} \alpha_s \boldsymbol{\tau}_s$ in Equation 4, we express $\boldsymbol{\theta}_{\text{merge}}(\boldsymbol{\theta})$ as $\boldsymbol{\theta}_0 + \sum_{s \neq t} \alpha_s \boldsymbol{\tau}_s + \alpha_t \boldsymbol{\tau}$, where $\boldsymbol{\tau} = \boldsymbol{\theta} - \boldsymbol{\theta}_0$, since $\boldsymbol{\theta}_t$ has not been found yet during the process of optimizing $\boldsymbol{\theta}$ for task $t$. We now have

$$\boldsymbol{\theta}_t = \arg \min_{\boldsymbol{\theta}} \mathcal{L}(\boldsymbol{\theta}_0 + \sum_{s \neq t} \alpha_s \boldsymbol{\tau}_s + \alpha_t \boldsymbol{\tau}; \mathcal{D}^{(t)})$$
$$= \arg \min_{\boldsymbol{\theta}} \mathcal{L}(\boldsymbol{\theta}_0 + \boldsymbol{\tau} - \boldsymbol{\tau} + \sum_{s \neq t} \alpha_s \boldsymbol{\tau}_s + \alpha_t \boldsymbol{\tau}; \mathcal{D}^{(t)})$$
$$= \arg \min_{\boldsymbol{\theta}} \mathcal{L}(\boldsymbol{\theta} - \boldsymbol{\tau} + \sum_{s \neq t} \alpha_s \boldsymbol{\tau}_s + \alpha_t \boldsymbol{\tau}; \mathcal{D}^{(t)}) \quad \because \boldsymbol{\theta} = \boldsymbol{\theta}_0 + \boldsymbol{\tau}$$
$$= \arg \min_{\boldsymbol{\theta}} \mathcal{L}(\boldsymbol{\theta} + \sum_{s \neq t} \alpha_s \boldsymbol{\tau}_s + (\alpha_t - 1) \boldsymbol{\tau}; \mathcal{D}^{(t)}).$$

## C    ADDITIONAL RESULTS

### C.1    EFFECT OF THE NUMBER OF STEPS DURING FINE-TUNING

SAFT enables continued accuracy gains through longer training without the risk of overfitting. Therefore, we conduct an ablation study to evaluate whether SAFT can enhance the performance of a single downstream task by doubling the number of training steps. In this study, we employ ASAM. As shown in Table C1, it appears that SGD converges, as performance plateaus after 2000 steps. In contrast, SAFT continues to consistently improve performance up to 8000 steps.

### C.2    FINE-TUNING PERFORMANCE OF SAFT VARIANTS

To empirically justify our choice of SAFT variants for our main experiments, we evaluate various the SAFT variants on the same datasets (i.e., eight vision tasks) as our main experiments. In particular,

Table C1: Average accuracies of fine-tuned ViT-B/32 over steps across the eight tasks.

| Fine-tuning steps | 2000 | 4000 | 8000 |
|---|---|---|---|
| SGD | 90.37 | 90.21 | 90.48 |
| SAFT-ASAM | **90.50** | **90.84** | **91.03** |

we investigate how the performance of a merged model changes when applying SAM (Foret et al., 2021), ASAM (Kwon et al., 2021), Friendly SAM (Li et al., 2024), WA-SAM (Kaddour et al., 2022), SAGM (Wang et al., 2023), PGN (Zhao et al., 2022), SSAM-F (Mi et al., 2022), and SSAM-D (Mi et al., 2022), as shown in Table C2. The results demonstrate that ASAM brings better performance improvement, compared to SAM and other SAFT variants. As a result, ASAM shows the best single-task performance among other variants. Therefore, we use ASAM as a default SAFT variant in all experiments.

Table C2: Average accuracy of fine-tuned ViT-B/32 over steps across various SAM variants and fine-tuning methods.

| SAFT variants | Accuracy |
|---|---|
| SGD | 90.45 |
| SAFT-SAM | 90.16 |
| SAFT-ASAM | **91.29** |
| SAFT-Friendly SAM | 90.29 |
| SAFT-WA-SAM | 91.06 |
| SAFT-SAGM | 90.96 |
| SAFT-PGN | 90.90 |
| SAFT-SSAM-F | 90.80 |
| SAFT-SSAM-D | 90.60 |

## C.3 CROSS-TASK LINEARITY (CTL)

We provide additional results that demonstrate that SAFT satisfies cross-task linearity on other pairs of datasets in Figure C1. We utilize ViT-B/32 as the image encoder to visualize this figure, just the same as Figure 3. The results show that our method achieves lower CTL scores across all layers for various task combinations. This suggests that our approach better satisfies CTL for a broader range of data, implying improved weight disentanglement and task arithmetic properties. Consequently, it can be concluded that our method reduces parameter interference and minimizes the performance gap.

## C.4 LOSS BETWEEN A MERGED MODEL AND TASK-SPECIFIC MODELS

To demonstrate that SAFT indeed reduces the loss sharpness and the performance gap between a merged model and task-specific models, we visualize loss changes as we traverse along a linear path between a merged model and a task-specific model on a given task in Figure C2. Sharpness-aware fine-tuning indeed results in reduced loss barrier, leading to less performance gap as exhibited in less weight disentanglement error, better cross-task linearity, and better overall performance in our main paper.

## C.5 ADDITIONAL RESULTS OF FINE-TUNING BASELINES AND MODEL MERGING METHODS

Following Section 6.2, we conduct experiments on all combinations of fine-tuning baselines (SGD, FTTS, FTLO) and model merging methods (weight averaging, task arithmetic, TIES), as summarized in Table C3. In the case of weight averaging, our method leads to performance improvements in most cases, and for task arithmetic, it achieves performance improvements in all cases. For weight averaging, our method improves performance in most cases, while task arithmetic consistently yields

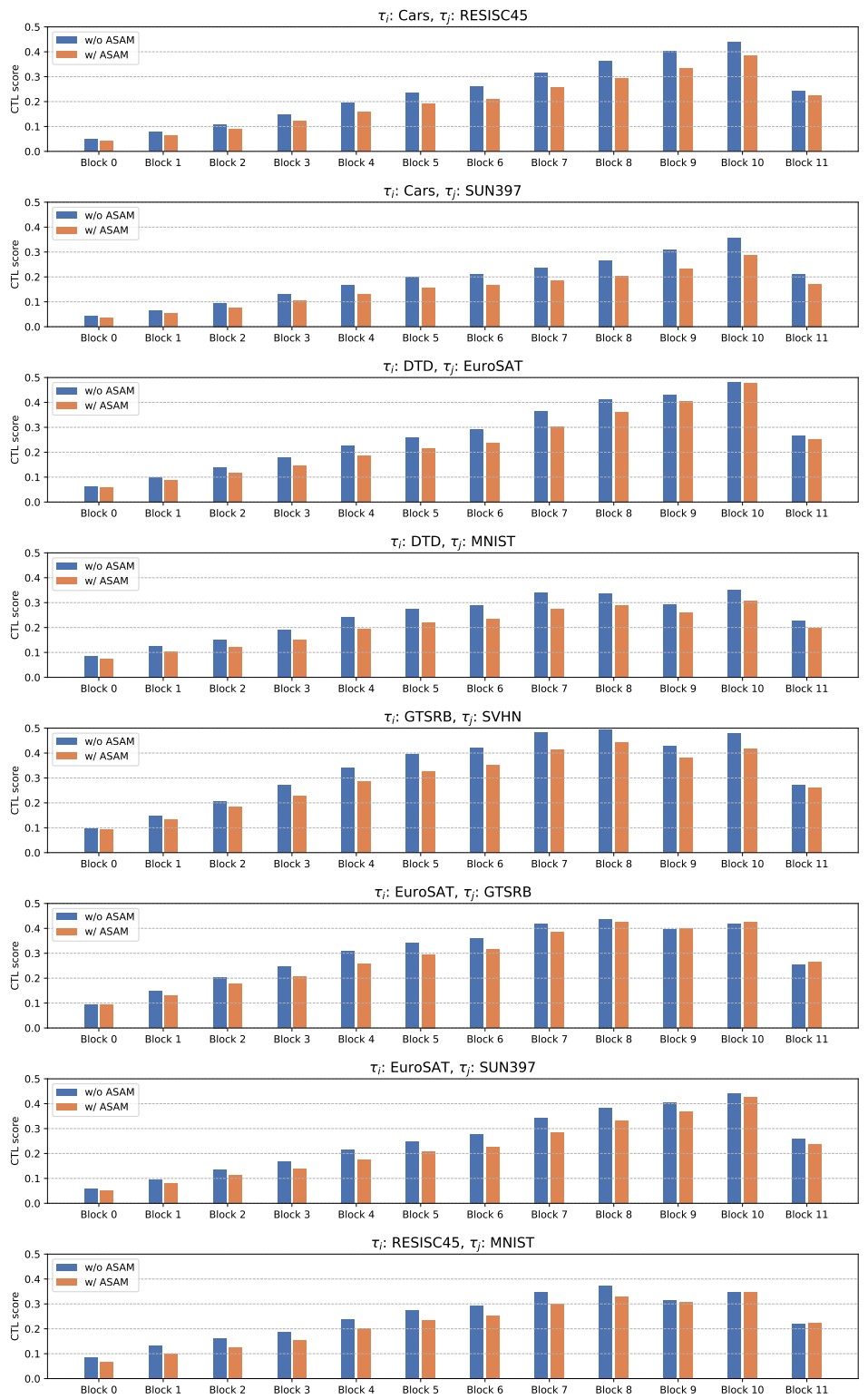

Figure C1: **Verification of all blocks CTL between the merged model and task-specific models**. We compare $\mathbb{E}_{\mathcal{D}^{(s)} \cup \mathcal{D}^{(t)}}[1 - \cos^{(\ell)}(\boldsymbol{x}; 2\lambda\boldsymbol{\tau}_s, 2\lambda\boldsymbol{\tau}_t)]$. We set the scaling factor $\lambda$ to 0.3.

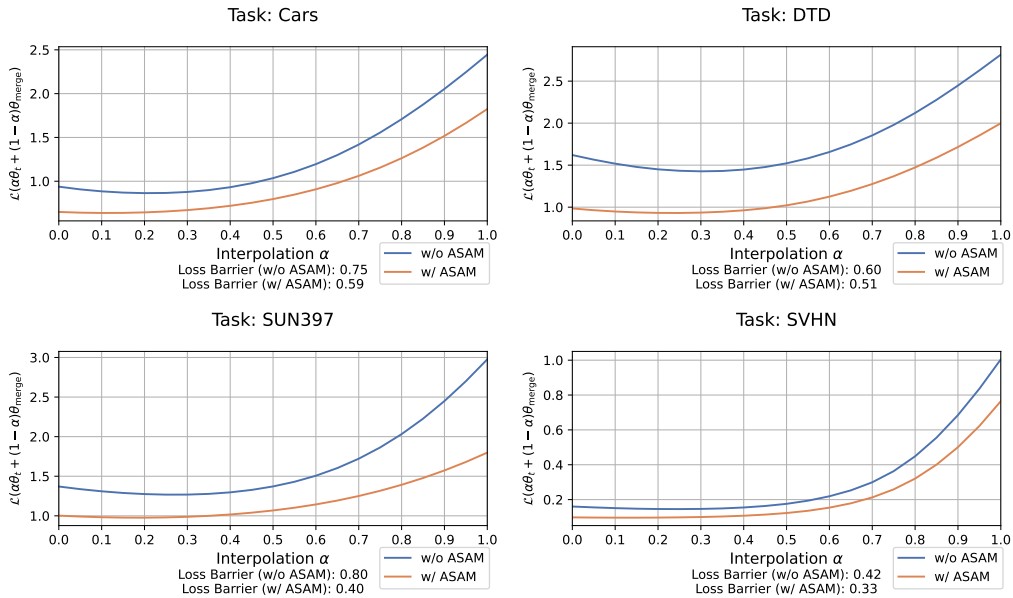

Figure C2: Test loss barrier between the merged model and each task-specific model.

performance improvements across all cases. In contrast, TIES shows performance improvements in only half of the cases. Upon closer examination, when linear fine-tuning methods such as FTTS and FTLO — which regularize the model output to satisfy linearity — are used without SAFT-ASAM, TIES generally outperforms task arithmetic. However, with SAFT-ASAM, TIES consistently performs worse than task arithmetic.

This seems that since the combination of linear fine-tuning and SAFT-ASAM has already enhanced weight disentanglement and reduced parameter interference, parameter trimming via TIES may rather remove critical parameters not noisy parameters, leading to performance degradation. Specifically, the combination of TIES and SAFT-ASAM delivers the best performance. Conversely, with FTTS and FTLO, task arithmetic paired with SAFT-ASAM achieves superior results. In some instances, TIES combined with SAFT-ASAM performs similarly to weight averaging. Thus, for linear fine-tuning methods like FTTS and FTLO, combining SAFT-ASAM with TIES can negatively impact performance. Additional analysis of this behavior is reserved for future work.

## C.6   MERGING WITH FIXED $\alpha_t$

Previous research (Ilharco et al., 2023; Jin et al., 2023; Matena & Raffel, 2022; Yadav et al., 2023) on model merging has focused on finding better merged models through hyperparameter tuning. However, such methods become increasingly costly as the number of hyperparameters grows, and they need to be re-applied whenever tasks are added or changed. Therefore, it is essential to create a robust merged model that performs well regardless of the selected hyperparameters.

Our method achieves robustness by identifying flatter minima for joint loss and weight disentanglement compared to SGD, enabling the discovery of optimal hyperparameters across a wider range of conditions. To support this claim, we evaluate task arithmetic by fixing the task coefficients $\alpha$ to $0.4$ for all merging tasks, following the recommendation of Ilharco et al. (2023). As shown in Table C4, our method outperforms other fine-tuning baselines in all cases, achieving improvements of up to 30% in both absolute accuracy and normalized accuracy. Additionally, there are several cases where the performances are nearly identical to those of hyperparameter tuning. Therefore, our method ensures that a merged model with reliable performance can be obtained, even when arbitrary hyperparameters are chosen.

Table C3: **Multi-task performance when merging a CLIP image encoder on eight tasks**. We report the average absolute and normalized accuracies for different five model merging methods. Results are shown for the six fine-tuning methods, categorized by whether SAFT-ASAM is applied.

| Merging methods (→) Fine-tuning baselines (↓) | Weight averaging | | Task arithmetic | | TIES merging | |
|---|---|---|---|---|---|---|
| | Abs. | Norm. | Abs. | Norm. | Abs. | Norm. |
| | ViT-B/32 | | | | | |
| SGD | 65.72 | 72.91 | 68.23 | 75.47 | 74.57 | 82.29 |
| SAFT-ASAM (Ours) | **66.76** | **73.62** | **69.45** | **76.32** | **75.45** | **82.86** |
| FTTS | 72.47 | 82.04 | 78.35 | 86.83 | **76.89** | **86.84** |
| FTTS w/ SAFT-ASAM (Ours) | **75.10** | **86.10** | **79.38** | **87.72** | 73.77 | 84.46 |
| FTLO | **65.96** | **73.83** | 75.93 | 85.74 | **77.39** | **85.89** |
| FTLO w/ SAFT-ASAM (Ours) | 65.34 | 72.78 | **77.49** | **88.77** | 76.30 | 84.62 |
| | ViT-B/16 | | | | | |
| SGD | 71.58 | 77.37 | 73.40 | 79.31 | 77.94 | 84.04 |
| SAFT-ASAM (Ours) | **71.84** | **77.53** | **76.77** | **82.50** | **80.14** | **86.23** |
| FTTS | 77.20 | 84.87 | 79.37 | 87.33 | **81.09** | **89.05** |
| FTTS w/ SAFT-ASAM (Ours) | **78.09** | **86.45** | **79.78** | **88.26** | 78.41 | 86.72 |
| FTLO | 70.97 | **77.11** | 80.00 | 86.55 | 78.25 | 84.91 |
| FTLO w/ SAFT-ASAM (Ours) | **71.03** | 76.78 | **82.59** | **89.11** | **79.49** | **85.92** |

Table C4: **Multi-task performance of task arithmetic with and without hyperparameter tuning**. We compare the average absolute and normalized accuracies of task arithmetic whether the hyperparameter is tuned. The fixed hyperparameter $\alpha$ is $0.4$ for all eight vision tasks.

| Fine-tuning baselines (↓) | w/o tuning | | w/ tuning | | w/o tuning | | w/ tuning | |
|---|---|---|---|---|---|---|---|---|
| | Abs. | Norm. | Abs. | Norm. | Abs. | Norm. | Abs. | Norm. |
| | ViT-B/32 | | | | ViT-B/16 | | | |
| SGD | 46.34 | 48.72 | 68.23 | 75.47 | 46.91 | 49.98 | 73.40 | 79.31 |
| SAFT-ASAM (Ours) | **57.27** | **62.06** | **69.45** | **76.32** | **71.37** | **76.34** | **76.77** | **82.50** |
| FTTS | 72.89 | 81.97 | 78.35 | 86.83 | 76.69 | 84.05 | 79.37 | 87.33 |
| FTTS w/ SAFT-ASAM (Ours) | **75.21** | **85.78** | **79.38** | **87.72** | **78.87** | **87.17** | **79.78** | **88.26** |
| FTLO | 48.20 | 55.29 | 75.93 | 85.74 | 77.36 | 83.60 | 80.00 | 86.55 |
| FTLO w/ SAFT-ASAM (Ours) | **79.68** | **87.92** | **77.49** | **88.77** | **82.50** | **88.95** | **82.59** | **89.11** |

## C.7 MULTI-TASK PERFORMANCE OF OTHER FLAT-MINIMA TECHNIQUES

We also evaluate the performance of other flat-minima techniques, in addition to SAFT variants like ASAM. Flat-minima techniques, including SWA (Izmailov et al., 2018), RWP (Li et al., 2022), and SAGM (Wang et al., 2023), are fine-tuning methods designed to minimize the loss while finding flat-minima during model training. As shown in Table C5, both our method and SWA improve multi-task performance compared to SGD. However, RWP and SAGM show worse performance than SGD. In addition, as discussed in Appendix C.6, we also present the results of performance evaluation without hyperparameter tuning in Table C5. Our method not only achieves the best performance when the hyperparameters are tuned but also outperforms in cases without tuning. Furthermore, the performance gap between our method and SGD, as well as other flat-minima techniques, widens significantly in these scenarios. This demonstrates that our method is superior to other flat-minima techniques and exhibits less performance degradation due to variations in parameter settings.

Table C5: **Multi-task performance across different flat-minima techniques**. We compare the average absolute and normalized accuracies of ViT-B/32 for five fine-tuning methods including three flat-minima techniques: SWA, RWP, and SAGM. We also compare the performance of merged model with and without hyperparameter tuning. All cases are merged with eight vision tasks by task arithmetic.

| Fine-tuning baselines ($\downarrow$) | w/o tuning | | w/ tuning | |
|---|---|---|---|---|
| | Abs. | Norm. | Abs. | Norm. |
| SGD | 46.34 | 48.72 | 68.23 | 75.47 |
| SWA | 48.94 | 52.46 | 68.58 | 76.11 |
| RWP | 34.97 | 36.58 | 62.48 | 73.02 |
| SAGM | 40.18 | 42.93 | 64.36 | 71.16 |
| SAFT-ASAM (Ours) | **57.27** | **62.06** | **69.45** | **76.32** |

The primary difference among flat-minima techniques lies in the strategies used to derive perturbations. This seems to influence how effectively these techniques can reduce the performance gap between each task-specific model and a merged model. Our method introduces perturbations by minimizing the loss difference between the current point in the parameter space and the point with the highest loss in its neighborhood during fine-tuning. This approach closely aligns with the objective of model merging, which aims to minimize the loss difference between the merged model and the individual task-specific models. Therefore, perturbation strategies derived from fine-tuning objectives similar to the model merging objective could result in greater performance improvements compared to other flat-minima techniques.

## C.8 RESULTS ON NATURAL LANGUAGE PROCESSING TASKS

Table C6: **Multi-task performance of the merged model across four NLP tasks**. We report the average absolute and normalized accuracies on four GLUE benchmark tasks: CoLA, MPRC, RTE, and SST-2. We fine-tune Flan-T5-base using either SGD or SAFT-ASAM and merge the four task-specific models.

| Fine-tuning baselines ($\downarrow$) | CoLA | | MRPC | | RTE | | SST-2 | | Average | |
|---|---|---|---|---|---|---|---|---|---|---|
| | Abs. | Norm. | Abs. | Norm. | Abs. | Norm. | Abs. | Norm. | Abs. | Norm. |
| SGD | 58.77 | 75.58 | 25.74 | 29.75 | 37.55 | 43.52 | 64.11 | 68.68 | 46.54 | 54.38 |
| FTTS | 66.06 | 92.61 | 28.19 | 34.96 | 1.81 | 2.35 | **87.39** | **94.90** | 45.86 | 56.21 |
| FTLO | 66.83 | 96.67 | **67.40** | **79.71** | 0 | 0 | 13.30 | 14.50 | 36.88 | 47.72 |
| SAFT-ASAM (Ours) | **68.65** | **99.31** | 42.16 | 52.92 | **49.82** | **60.00** | 45.30 | 49.38 | **51.48** | **75.47** |

To demonstrate the effectiveness of our method in other domains, we evaluate our method on NLP tasks. Following the evaluation settings of Ilharco et al. (2023), we fine-tune the Flan-T5-base (Raffel et al., 2019; Wei et al., 2022) on four natural language processing (NLP) tasks: CoLA, MPRC, RTE, and SST-2 in GLUE benchmark (Wang et al., 2019). All fine-tuning processes start from the Flan-T5-base pre-trained checkpoint available on HuggingFace. We fine-tune each model for 8000 iterations with a batch size of 16 and a learning rate of $10^{-5}$. AdamW is used as the optimizer, and a linear annealing approach without warmup is applied as the learning rate scheduler. For efficient fine-tuning, we convert all downstream NLP tasks into a text-to-text format, following the approach in Jin et al. (2025). We measure the multi-task performance of the multi-task model merged by all four tasks using task arithmetic.

Table C6 indicates that FTTS and FTLO exhibit a tendency for certain finetuned parameters to exert undue influence after the merging process. This dominance leads to significant performance degradation on specific tasks, notably RTE, where both FTTS and FTLO experience extreme performance drops. These results suggest that FTTS and FTLO are still susceptible to parameter interference.

Although they achieve commendable performance on some tasks, their performance is inconsistent across the entire task set. In contrast, our proposed method effectively mitigates parameter interference, resulting in a more balanced performance profile across all tasks. This balanced approach leads to a superior average performance, as demonstrated by the highest absolute and normalized accuracy, with the normalized accuracy being approximately 20% higher than the second-best performing method, FTTS.

The parameter dominance observed in FTTS and FTLO is further substantiated by an analysis of the predicted text from the merged model. For clarity, the label texts for CoLA are "unacceptable" or "acceptable", for MRPC "not_equivalent" or "equivalent", for RTE "entailment" or "not_entailment", and for SST-2 "negative" or "positive". This label text configuration aligns with the settings established in Jin et al. (2025). Our analysis reveals that texts predicted by FTLO frequently contain or resemble "acceptable", "unacceptable", "equivalent", or "not_equivalent", irrespective of the input text data. While these outputs may be appropriate for CoLA and MRPC datasets, they are irrelevant for RTE and SST-2, resulting in the observed performance decline on these tasks. This analysis highlights that after merging, FTTS is significantly influenced by the parameters finetuned for specific tasks like SST-2 and CoLA. FTLO exhibits a similar pattern of influence. Conversely, although our predictions of our method may not always be correct, it consistently predicts one of the correct text labels for the corresponding task of the input data, across all four tasks. This consistency demonstrates an improved ability to mitigate parameter interference. Notably, while FTLO and FTTS outperform our method on only one task each, our method achieves superior performance on the remaining three tasks. Specifically, our method outperforms FTLO on CoLA, RTE, and SST-2, while also outperforming FTTS on CoLA, MRPC, and RTE. This result demonstrates an improvement in overall performance and suggests a greater resilience to parameter interference.

### C.9  TRAINING COSTS OF FINE-TUNING

Table C7: Training cost of SAFT.

| Fine-tuning baselines ($\downarrow$) | ViT-B/32 | | ViT-B/16 | |
|---|---|---|---|---|
| | Time (it/s) | VRAM (GB) | Time (it/s) | VRAM (GB) |
| SGD | 2.53 | 7.3 | 0.68 | 21.5 |
| FTTS | 1.30 | 12.6 | 0.37 | 37.2 |
| FTLO | 3.11 | 5.8 | 0.84 | 19.0 |
| SAFT-ASAM (Ours) | 1.34 | 7.8 | 0.34 | 21.6 |

Table C7 presents a comparison of training costs between SGD and our method across various models and fine-tuning methods.

We use AdamW as an optimizer, setting the batch size to 128. Additionally, all training is conducted using NVIDIA Quadro RTX 8000 GPUs.

The use of SAFT-ASAM leads to a nearly twofold increase in training time, whereas VRAM usage experiences only a minor increment.

Recently, there has been active research aimed at reducing the computational cost of SAM (Du et al., 2022; Liu et al., 2022). Model merging is an approach designed to efficiently build multi-task models, and since our work seeks to establish a connection between model merging and SAFT, we believe our research can significantly contribute to works focused on improving the efficiency of SAFT.

## D  PROOF FOR THEOREM 1

In this section, we provide a proof for Theorem 1, which is restated below for the convenience:

**Theorem 1** (SAFT induces joint-task loss linearity). *If models, parameterized by $\theta_s$ and $\theta_t$, are obtained by fine-tuning from a common pre-trained model via SAFT on their respective datasets*

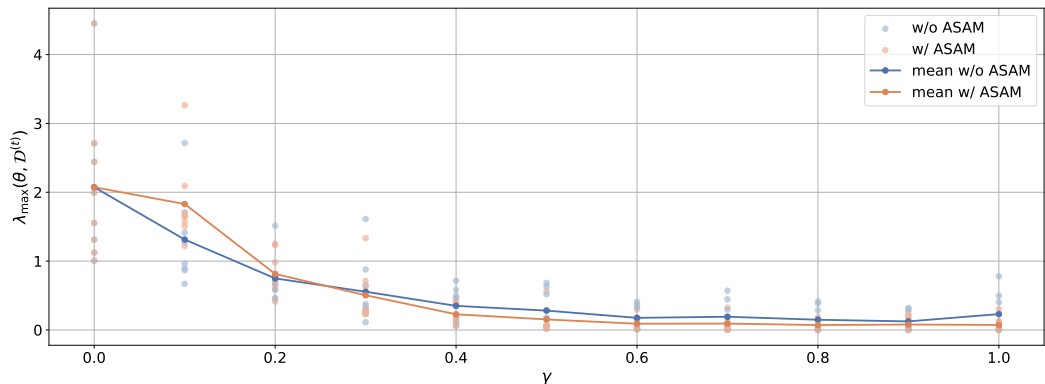

Figure D1: **Comparison of the dominant Hessian eigenvalue for parameters on the line segment between the pre-trained parameters and fine-tuned parameters.** We compare the dominant Hessian eigenvalue $\lambda_{\max}(\boldsymbol{\theta}; \mathcal{D}^{(t)})$ of parameter $\boldsymbol{\theta}$ along the line segment $\overline{\boldsymbol{\theta}_0\boldsymbol{\theta}_t}$, where $\boldsymbol{\theta} = \boldsymbol{\theta}_0 + \gamma(\boldsymbol{\theta}_t - \boldsymbol{\theta}_0)$ for $\gamma \in [0, 1]$, $\mathcal{D}^{(t)}$ denotes the dataset for task k. The line represents the mean of the dominant Hessian eigenvalues for all tasks.

$\mathcal{D}_s$ and $\mathcal{D}_t$, *the models are linearly connected on the loss landscape over the joint-task datasets* ($\mathcal{D} = \mathcal{D}_s \cup \mathcal{D}_t$).

*Proof.* Let $\delta$ represent the difference between the linear interpolation of Joint-Task Losses incurred by task-specific models parameterized by $\boldsymbol{\theta}_s$ and $\boldsymbol{\theta}_t$ and Joint-Task Loss incurred by a interpolated model parameterized by $\alpha\boldsymbol{\theta}_s + (1 - \alpha)\boldsymbol{\theta}_t$:

$$\delta = \mathcal{L}_{JTL}(\alpha\boldsymbol{\theta}_s + (1 - \alpha)\boldsymbol{\theta}_t; \mathcal{D}) - \alpha\mathcal{L}_{JTL}(\boldsymbol{\theta}_s; \mathcal{D}) - (1 - \alpha)\mathcal{L}_{JTL}(\boldsymbol{\theta}_t; \mathcal{D}). \tag{15}$$

$$
\begin{aligned}
\delta &= \mathcal{L}_{JTL}(\alpha\boldsymbol{\theta}_s + (1 - \alpha)\boldsymbol{\theta}_t; \mathcal{D}) - \alpha\mathcal{L}_{JTL}(\boldsymbol{\theta}_s; \mathcal{D}) - (1 - \alpha)\mathcal{L}_{JTL}(\boldsymbol{\theta}_t; \mathcal{D}) \\
&= [\mathcal{L}(\alpha\boldsymbol{\theta}_s + (1 - \alpha)\boldsymbol{\theta}_t; \mathcal{D}_s) - \alpha\mathcal{L}(\boldsymbol{\theta}_s; \mathcal{D}_s) - (1 - \alpha)\mathcal{L}(\boldsymbol{\theta}_t; \mathcal{D}_s)] \\
&\quad + [\mathcal{L}(\alpha\boldsymbol{\theta}_s + (1 - \alpha)\boldsymbol{\theta}_t; \mathcal{D}_t) - \alpha\mathcal{L}(\boldsymbol{\theta}_s; \mathcal{D}_t) - (1 - \alpha)\mathcal{L}(\boldsymbol{\theta}_t; \mathcal{D}_t)] \\
&= \delta_s + \delta_t, \\
where \quad \delta_s &= \mathcal{L}(\alpha\boldsymbol{\theta}_s + (1 - \alpha)\boldsymbol{\theta}_t; \mathcal{D}_s) - \alpha\mathcal{L}(\boldsymbol{\theta}_s; \mathcal{D}_s) - (1 - \alpha)\mathcal{L}(\boldsymbol{\theta}_t; \mathcal{D}_s), \\
\delta_t &= \mathcal{L}(\alpha\boldsymbol{\theta}_s + (1 - \alpha)\boldsymbol{\theta}_t; \mathcal{D}_t) - \alpha\mathcal{L}(\boldsymbol{\theta}_s; \mathcal{D}_t) - (1 - \alpha)\mathcal{L}(\boldsymbol{\theta}_t; \mathcal{D}_t).
\end{aligned}
\tag{16}
$$

Performing a third-order Taylor expansion of $\mathcal{L}(\alpha\boldsymbol{\theta}_s + (1 - \alpha)\boldsymbol{\theta}_t; \mathcal{D}_s)$ around $\boldsymbol{\theta}_s$:

$$
\begin{aligned}
\mathcal{L}(\alpha\boldsymbol{\theta}_s + (1 - \alpha)\boldsymbol{\theta}_t; \mathcal{D}_s) &= \mathcal{L}(\boldsymbol{\theta}_s; \mathcal{D}_s) + (1 - \alpha)\nabla_{\boldsymbol{\theta}}\mathcal{L}(\boldsymbol{\theta}_s; \mathcal{D}_s)^\top(\boldsymbol{\theta}_t - \boldsymbol{\theta}_s) \\
&\quad + \frac{1}{2}(1 - \alpha)^2(\boldsymbol{\theta}_t - \boldsymbol{\theta}_s)^\top\mathbf{H}_s(\boldsymbol{\theta}_t - \boldsymbol{\theta}_s) + R_s,
\end{aligned}
\tag{17}
$$

where $\mathbf{H}_s = \nabla_{\boldsymbol{\theta}}^2\mathcal{L}(\boldsymbol{\theta}_s; \mathcal{D}_s)$ and $R_s$ is the remainder term.

Similarly, expand $\mathcal{L}(\boldsymbol{\theta}_t; \mathcal{D}_s)$ around $\boldsymbol{\theta}_s$:

$$\mathcal{L}(\boldsymbol{\theta}_t; \mathcal{D}_s) = \mathcal{L}(\boldsymbol{\theta}_s; \mathcal{D}_s) + \nabla_{\boldsymbol{\theta}}\mathcal{L}(\boldsymbol{\theta}_s; \mathcal{D}_s)^\top(\boldsymbol{\theta}_t - \boldsymbol{\theta}_s) + \frac{1}{2}(\boldsymbol{\theta}_t - \boldsymbol{\theta}_s)^\top\mathbf{H}_s(\boldsymbol{\theta}_t - \boldsymbol{\theta}_s) + R'_s. \tag{18}$$

Multiply both sides by $(1 - \alpha)$:

$$
\begin{aligned}
(1 - \alpha)\mathcal{L}(\boldsymbol{\theta}_t; \mathcal{D}_s) &= (1 - \alpha)\mathcal{L}(\boldsymbol{\theta}_s; \mathcal{D}_s) + (1 - \alpha)\nabla_{\boldsymbol{\theta}}\mathcal{L}(\boldsymbol{\theta}_s; \mathcal{D}_s)^\top(\boldsymbol{\theta}_t - \boldsymbol{\theta}_s) \\
&\quad + \frac{1}{2}(1 - \alpha)(\boldsymbol{\theta}_t - \boldsymbol{\theta}_s)^\top\mathbf{H}_s(\boldsymbol{\theta}_t - \boldsymbol{\theta}_s) + (1 - \alpha)R'_s.
\end{aligned}
\tag{19}
$$

Compute $\delta_s$:

$$
\begin{aligned}
\delta_s &= \mathcal{L}(\alpha\boldsymbol{\theta}_s + (1-\alpha)\boldsymbol{\theta}_t; \mathcal{D}_s) - \alpha\mathcal{L}(\boldsymbol{\theta}_s; \mathcal{D}_s) - (1-\alpha)\mathcal{L}(\boldsymbol{\theta}_t; \mathcal{D}_s) \\
&= \left[ \mathcal{L}(\boldsymbol{\theta}_s; \mathcal{D}_s) + (1-\alpha)\nabla_{\boldsymbol{\theta}}\mathcal{L}(\boldsymbol{\theta}_s; \mathcal{D}_s)^\top(\boldsymbol{\theta}_t - \boldsymbol{\theta}_s) + \frac{1}{2}(1-\alpha)^2(\boldsymbol{\theta}_t - \boldsymbol{\theta}_s)^\top\mathbf{H}_s(\boldsymbol{\theta}_t - \boldsymbol{\theta}_s) + R_s \right] \\
&\quad - \alpha\mathcal{L}(\boldsymbol{\theta}_s; \mathcal{D}_s) - \Big[ (1-\alpha)\mathcal{L}(\boldsymbol{\theta}_s; \mathcal{D}_s) + (1-\alpha)\nabla_{\boldsymbol{\theta}}\mathcal{L}(\boldsymbol{\theta}_s; \mathcal{D}_s)^\top(\boldsymbol{\theta}_t - \boldsymbol{\theta}_s) \\
&\quad + \frac{1}{2}(1-\alpha)(\boldsymbol{\theta}_t - \boldsymbol{\theta}_s)^\top\mathbf{H}_s(\boldsymbol{\theta}_t - \boldsymbol{\theta}_s) + (1-\alpha)R'_s \Big].
\end{aligned}
\tag{20}
$$

Simplify the expression:

$$
\begin{aligned}
\delta_s &= \mathcal{L}(\boldsymbol{\theta}_s; \mathcal{D}_s) - \alpha\mathcal{L}(\boldsymbol{\theta}_s; \mathcal{D}_s) - (1-\alpha)\mathcal{L}(\boldsymbol{\theta}_s; \mathcal{D}_s) \\
&\quad + (1-\alpha)\nabla_{\boldsymbol{\theta}}\mathcal{L}(\boldsymbol{\theta}_s; \mathcal{D}_s)^\top(\boldsymbol{\theta}_t - \boldsymbol{\theta}_s) - (1-\alpha)\nabla_{\boldsymbol{\theta}}\mathcal{L}(\boldsymbol{\theta}_s; \mathcal{D}_s)^\top(\boldsymbol{\theta}_t - \boldsymbol{\theta}_s) \\
&\quad + \frac{1}{2}(1-\alpha)^2(\boldsymbol{\theta}_t - \boldsymbol{\theta}_s)^\top\mathbf{H}_s(\boldsymbol{\theta}_t - \boldsymbol{\theta}_s) - \frac{1}{2}(1-\alpha)(\boldsymbol{\theta}_t - \boldsymbol{\theta}_s)^\top\mathbf{H}_s(\boldsymbol{\theta}_t - \boldsymbol{\theta}_s) \\
&\quad + R_s - (1-\alpha)R'_s. \\
&= -\frac{1}{2}\alpha(1-\alpha)(\boldsymbol{\theta}_t - \boldsymbol{\theta}_s)^\top\mathbf{H}_s(\boldsymbol{\theta}_t - \boldsymbol{\theta}_s) + (R_s - (1-\alpha)R'_s).
\end{aligned}
\tag{21}
$$

Similarly, compute $\delta_t$ by expanding around $\boldsymbol{\theta}_t$:

$$
\delta_t = -\frac{1}{2}\alpha(1-\alpha)(\boldsymbol{\theta}_t - \boldsymbol{\theta}_s)^\top\mathbf{H}_t(\boldsymbol{\theta}_t - \boldsymbol{\theta}_s) + (R_t - \alpha R'_t),
$$
$$
\text{where} \quad \mathbf{H}_t = \nabla^2_{\boldsymbol{\theta}}\mathcal{L}(\boldsymbol{\theta}_t; \mathcal{D}_t).
\tag{22}
$$

Combining $\delta_s$ and $\delta_t$:

$$
\delta = \delta_s + \delta_t = -\frac{1}{2}\alpha(1-\alpha)(\boldsymbol{\theta}_t - \boldsymbol{\theta}_s)^\top(\mathbf{H}_s + \mathbf{H}_t)(\boldsymbol{\theta}_t - \boldsymbol{\theta}_s) + \epsilon,
$$
$$
\text{where} \quad \epsilon = (R_s - (1-\alpha)R'_s) + (R_t - \alpha R'_t).
\tag{23}
$$

Since the dominant Hessian eigenvalue $\lambda_{\max}(\boldsymbol{\theta}; \mathcal{D})$ is the largest eigenvalue of $\mathbf{H}(\boldsymbol{\theta}; \mathcal{D})$, we have

$$
\begin{aligned}
(\boldsymbol{\theta}_t - \boldsymbol{\theta}_s)^\top\mathbf{H}_s(\boldsymbol{\theta}_t - \boldsymbol{\theta}_s) &\le \lambda_s\|\boldsymbol{\theta}_t - \boldsymbol{\theta}_s\|^2, \\
(\boldsymbol{\theta}_t - \boldsymbol{\theta}_s)^\top\mathbf{H}_t(\boldsymbol{\theta}_t - \boldsymbol{\theta}_s) &\le \lambda_t\|\boldsymbol{\theta}_t - \boldsymbol{\theta}_s\|^2,
\end{aligned}
\tag{24}
$$

where $\lambda_s = \lambda_{\max}(\boldsymbol{\theta}_s; \mathcal{D}_s), \lambda_t = \lambda_{\max}(\boldsymbol{\theta}_t; \mathcal{D}_t)$. Then, $|\delta|$ is bounded as:

$$
|\delta| \le \frac{1}{2}\alpha(1-\alpha)(\lambda_s + \lambda_t)\|\boldsymbol{\theta}_t - \boldsymbol{\theta}_s\|^2 + \epsilon,
\tag{25}
$$

where $\epsilon = (R_s - (1-\alpha)R'_s) + (R_t - \alpha R'_t)$ is the remainder term. Agarwala & Dauphin (2023) have demonstrated that SAM reduces the dominant Hessian eigenvalue throughout the learning trajectory. According to Equation 25, the reduction of the dominant Hessian eigenvalue by SAFT leads to smaller $|\delta|$ and thus better satisfaction of joint-task loss linearity. $\qquad\square$

