# OpenReview forum: "Mitigating Parameter Interference in Model Merging via Sharpness-Aware Fine-Tuning"
_ICLR.cc/2025/Conference — ICLR 2025 Poster_

### Official Review · Reviewer_cmQk · 2024-11-01

**Soundness:** 3
**Presentation:** 3
**Contribution:** 3
**Rating:** 6
**Confidence:** 4

**Summary:**

This paper proposes leveraging Sharpness-Aware Minimization (SAM) during fine-tuning to enhance the performance of model merging. The authors find that SAM effectively reduces parameter interference, addressing a core challenge in model merging. With its inherent capacity to improve generalization, this paper shows that SAM can be a viable and effective optimization method for model merging.

**Strengths:**

This paper presents an interesting perspective on improving model merging by using SAM as the optimizer. The authors show convincing evidence that SAM can reduce parameter interference, which has not been deeply explored. The experiment results also show improvement of SAM in many scenarios.

**Weaknesses:**

1. The experiments in Table 2 rely on SGD as the base optimizer, whereas Table 1 indicates that FTTS and FTLO significantly outperform SGD. It would be beneficial to show results combining FTTS, FTLO, and TIES merging to better understand the method's upper performance bounds.
2. The experimental scope is limited to vision tasks, and expanding the evaluation to include NLP tasks would strengthen the demonstration of the method’s applicability.

**Questions:**

Please refer to the weaknesses highlighted above

---

### Official Review · Reviewer_22A4 · 2024-11-02

**Soundness:** 3
**Presentation:** 3
**Contribution:** 2
**Rating:** 6
**Confidence:** 3

**Summary:**

The paper presents a method that aims to reduce the interference during the merging of multiple models. This is achieved by optimizing the performance gap between the merged model and each individually finetuned model, as well as optimizing per-task losses. To minimize these objectives, the authors incorporate Sharpness-Aware Minimization (SAM) during the finetuning process. This approach not only helps reduce parameter interference but also enhances the generalization of finetuned models. Empirical results suggest that SAM facilitates weight disentanglement and improves cross-task linearity. Additionally, the performance of the final model is enhanced on different merging methods, which demonstrates the orthogonality of the proposed method.

**Strengths:**

The paper is well-structured and articulate, making it easy to follow. It presents an intriguing finding that employing SAM can significantly narrow the performance gap between a merged model and task-specific models. The experiments conducted across various merging methods effectively illustrate the effectiveness of the proposed approach.

**Weaknesses:**

The connection between the objective of SAM and Equation (7) is relatively loose and it is uclear how SAM can help minimize the objective in Equation (7). Thus, further ablation studies are needed to demonstrate the motivation of SAM. Otherwise, it seems that any optimization technique that targets flat minima could potentially enhance the performance of the merged model by steering the parameters towards regions where interpolation between different parameters does not increase the loss values.

**Questions:**

- Could you clarify how the parameters $\alpha_t$ with $t>2$ are defined in Equations (10) and (12)?
- I am wondering how other flat minima techniques can help bridge the performance gap between merged and individually finetuned models (e.g. SWA [1], RWP [2], ...).
- It would be beneficial to discuss a related work [3], which aims to find the common low and flat loss region of per-task objectives.

[1] Izmailov, Pavel, et al. "Averaging weights leads to wider optima and better generalization." arXiv preprint arXiv:1803.05407 (2018).

[2] Li, Tao, et al. "Efficient generalization improvement guided by random weight perturbation." arXiv preprint arXiv:2211.11489 (2022).

[3] Phan, Hoang, et al. "Improving multi-task learning via seeking task-based flat regions." arXiv preprint arXiv:2211.13723 (2022).

---

### Official Review · Reviewer_AjRq · 2024-11-03

**Soundness:** 3
**Presentation:** 2
**Contribution:** 2
**Rating:** 6
**Confidence:** 4

**Summary:**

This paper propose utilizing Sharpness-Aware Minimization (SAM) during the finetuning of pre-trained models to achieve better generalization and weight disentanglement for model merging. The central hypothesis is that SAM can lead to flatter minima, which in turn reduces interference between task-specific models and enhances the performance of the merged model. The authors demonstrate the effectiveness through several experiments.

**Strengths:**

* The description of the background is very clear, making the motivation of this work reasonable.
* The paper shows consistent performance improvements in merged models.
* They demonstrate the effectiveness of this method through weight disentanglement visualization and CTL.

**Weaknesses:**

* The contributions seem to be not enough.
* Limited experimental results in this paper. They focus on the vision tasks and the results on other domains are unclear.
* The performance gain of several results are relatively small.

**Questions:**

* The resource cost of this method should be reported. Does this method increases the training cost than the baseline?

---

### Official Review · Reviewer_ENRa · 2024-11-04

**Soundness:** 2
**Presentation:** 3
**Contribution:** 2
**Rating:** 5
**Confidence:** 3

**Summary:**

This paper presents a model merging method by fine-tuning each pre-trained model using SAM, aiming to reduce parameter interference and improve task-specific performance. A comprehensive empirical analysis in weight disentanglement, along with experiments, demonstrates the effectiveness of the proposed methods.

**Strengths:**

1. This paper is well-written and well-organized. I enjoyed reading it.
2. The analysis in Section 5 is comprehensive, although primarily from an empirical perspective.
3. The results are promising for some merging methods.
4. The method is versatile can be applied to existing merging approaches.

**Weaknesses:**

1. Confused motivation: For Eq.(6), the proposed method still needs to use the dataset of each task and fine-tune $\theta_t$ of each task. Therefore, if we have all task-specific datasets, why not directly perform joint training?

2. Using "sharpness-aware" is misleading for the goal of addressing parameter interference. Although Eq.(7) takes a similar form to the SAM function Eq.(2), the perturbations in Eq.(7) are not derived from the goal of "sharpness-awareness." In my opinion, there is no link between "sharpness-aware" and "interference reduction."

3. When optimizing Eq.(6), we optimize $\theta_t$ while keeping other models frozen. Afterwards, when optimizing $\theta_s$, which version of $\theta_t$ should we use? If we use the latest version, which task-specific model should we optimize first? Should we consider the forgetting issue for this sequential optimization?

**Questions:**

1. Could you interpret the meaning of the perturbations in Eq.(7)? This could provide readers with more intuitive explanations.

---

### Meta-Review · Area_Chair_AcQ1 · 2024-12-21

**Metareview:**

The paper proposes a method to perform model model merging by changing the loss function during training so that it makes the model merging process easier later. This paper leverages Sharpness-Aware Minimization (SAM) during fine-tuning to reduce parameter interference and improve task-specific performance. SAM promotes flatter minima, enhancing generalization and facilitating weight disentanglement, which addresses key challenges in merging multiple models. Reviewers agree that the paper is well written, easy to follow and the proposed method is novel. The initial weakness were attributed to explaining the SAM loss as well as evaluations. The authors presented extensive that reviewers found useful. Hence based on the final ratings, the paper can be accepted to ICLR

**Additional Comments On Reviewer Discussion:**

The authors presented thorough rebuttals and every reviewers actively participated in the rebuttal process.

---

### Decision · Program_Chairs · 2025-01-22

Accept (Poster)